# Inhibitory Effects of Loganin on Adipogenesis In Vitro and In Vivo

**DOI:** 10.3390/ijms24054752

**Published:** 2023-03-01

**Authors:** Hyoju Jeon, Chang-Gun Lee, Hyesoo Jeong, Seong-Hoon Yun, Jeonghyun Kim, Laxmi Prasad Uprety, Kang-Il Oh, Shivani Singh, Jisu Yoo, Eunkuk Park, Seon-Yong Jeong

**Affiliations:** 1Department of Medical Genetics, Ajou University School of Medicine, Suwon 16499, Republic of Korea; 2Department of Biomedical Sciences, Ajou University School of Medicine, Suwon 16499, Republic of Korea; 3Nine B Co., Ltd., Daejeon 34121, Republic of Korea

**Keywords:** loganin, anti-adipogenic effect, ovariectomized mice, high-fat diet mice

## Abstract

Obesity is characterized by the excessive accumulation of mature adipocytes that store surplus energy in the form of lipids. In this study, we investigated the inhibitory effects of loganin on adipogenesis in mouse preadipocyte 3T3-L1 cells and primary cultured adipose-derived stem cells (ADSCs) in vitro and in mice with ovariectomy (OVX)- and high-fat diet (HFD)-induced obesity in vivo. For an in vitro study, loganin was co-incubated during adipogenesis in both 3T3-L1 cells and ADSCs, lipid droplets were evaluated by oil red O staining, and adipogenesis-related factors were assessed by qRT-PCR. For in vivo studies, mouse models of OVX- and HFD-induced obesity were orally administered with loganin, body weight was measured, and hepatic steatosis and development of excessive fat were evaluated by histological analysis. Loganin treatment reduced adipocyte differentiation by accumulating lipid droplets through the downregulation of adipogenesis-related factors, including peroxisome proliferator-activated receptor γ (*Pparg*), CCAAT/enhancer-binding protein α (*Cebpa*), perilipin 2 (*Plin2*), fatty acid synthase (*Fasn*), and sterol regulatory element binding transcription protein 1 (*Srebp1*). Loganin administration prevented weight gain in mouse models of obesity induced by OVX and HFD. Further, loganin inhibited metabolic abnormalities, such as hepatic steatosis and adipocyte enlargement, and increased the serum levels of leptin and insulin in both OVX- and HFD-induced obesity models. These results suggest that loganin is a potential candidate for preventing and treating obesity.

## 1. Introduction

Obesity is a crucial health problem worldwide, and it is caused by hormonal abnormalities, genetic factors, and an imbalance between food intake and energy consumption [1]. Body mass index (BMI), calculated by dividing body weight by the square of height, is the most commonly used diagnostic indicator of obesity [2]. According to the World Health Organization (WHO) guidelines, a BMI of 25–30 and > 30 kg/m^2^ are considered overweight and obese, respectively [3]. In 2016, 1.9 billion and 650 million adults above 18 years of age were reported to be overweight and obese, respectively [4].

Obesity is characterized by the abnormal deposition of fat in the body, leading to metabolic abnormalities, including fatty liver, elevated plasma insulin/leptin levels, and dyslipidemia [5]. Liver steatosis is caused by an increase in liver fat, which can promote inflammatory signaling pathways that trigger oxidative stress in hepatocytes and produce proinflammatory cytokine. This can lead to the development of non-alcoholic steatohepatitis and macrophage infiltration, which cause liver damage [6,7]. Moreover, excessive fat accumulation alters two main endocrine factors: insulin and leptin [8]. Insulin is a hormone secreted from pancreatic β cells when large amounts of energy are consumed. Insulin regulates energy metabolism by converting glucose into fat. In obese individuals, elevated plasma insulin levels have been observed, in which insulin sensitivity is reduced in insulin-targeted organs such as the liver and adipose tissues, which results in excessive insulin production [9]. Excessive differentiated adipocytes trigger excessive fat accumulation, which leads to an increase in the number or the size of adipocytes (hypertrophy), resulting in a high risk of obesity [10,11].

Adipogenesis is a process in which surplus energy is stored in adipocytes in the form of lipids [12]. Adipogenesis is the process of differentiation of mesenchymal stem cells (MSCs) into adipocytes [13]. MSCs are differentiated by a complex cascade of adipocyte-specific transcription factors, such as peroxisome proliferator-activated receptor γ (*Ppar*g), CCAAT/enhancer-binding protein α (*Cebp*a), perilipin 2 (*Plin2*), fatty acid synthase (*Fasn*), and sterol regulatory element binding transcription protein 1 (*Srebp1*) [14,15,16]. These genes are essential adipogenesis-related markers regulating adipocyte differentiation [15,16]. Excessive differentiated adipocytes trigger immoderate fat accumulation, which leads to an increase in the number of adipocytes (hyperplasia) or the size of adipocytes (hypertrophy), resulting in a high risk of obesity [17]. Despite having a relatively short life in plasma, adipocytokines such as leptin and adiponectin play a crucial role in regulating fat accumulation, which influences insulin sensitivity [18].

Overweightness is generally caused by abnormal eating behavior (i.e., calorie-rich food intake, irregular eating habits, and snacking after a meal), insufficient exercise, and inadequate sleep time [19]. Recently, pharmacological therapies, including liraglutide (suppressing appetite) and orlistat (decreasing fat absorption) for managing and preventing obesity, have seen an increase in patients with obesity. However, some medications have serious adverse effects and long-term safety limitations, such as vomiting, nausea, satiety, and oily evacuation [20].

Medicinal herbs have been widely considered as alternative conventional therapeutics in the treatment and prevention of various diseases, owing to their long-term safety and fewer adverse effects [21]. A study has demonstrated that single bioactive components derived from herbal products have beneficial therapeutic effects as natural medicines [22]. In addition, studies have shown that several plants containing the iridoid glycoside bioactive compound loganin alleviated hepatic steatosis in a non-alcoholic fatty liver disease mouse model [23], exhibit antidiabetic activities in obese diabetic rats [24] and inhibit adipocyte differentiation and proliferation in rat preadipocytes [25]. Further, loganin prevents inflammatory responses in mouse 3T3-L1 preadipocyte cells and in Tyloxapol-induced mice [26] resulting in decreased body weight gain via improved glucolipid metabolism [25]. Although several beneficial effects of loganin are known, the specific anti-obesity effects of loganin on adipogenesis remain unclear.

Therefore, this study aimed to investigate the inhibitory effects of loganin in 3T3-L1 mouse preadipocytes and adipose-derived stem cells (ADSCs) in vitro and in ovariectomy (OVX) and high-fat diet (HFD)-induced mice in vivo.

## 2. Results

### 2.1. Loganin Inhibits Adipocyte Differentiation in Mouse Preadipocytes and ADSCs

We first examined whether loganin inhibits adipogenesis in 3T3-L1 mouse preadipocyte cells. Cells were induced to differentiate into adipocytes and were co-incubated with different concentrations of loganin (2, 5, and 10 μM) for 8 d. After the induction of adipocyte differentiation, mRNA expression levels of adipogenic-related markers such as Pparg and Cebpa for adipogenesis, Plin2 for mature adipocytes, and Fasn and Srebp1 for upstream activator of adipogenesis were examined using quantitative reverse transcription polymerase chain reaction (qRT-PCR), and accumulated lipid droplets were visualized using oil Red O staining. Loganin significantly decreased the mRNA expression levels of Pparg, Cebpa, Plin2, Fasn, and Srebp1 in a dose-dependent manner, and treatment with 10 μM loganin showed the greatest inhibitory effect on adipocyte differentiation (Figure 1A). Loganin treatment decreased the number of oil Red O-positive cells (Figure 1B). Further, the cellular viability test showed that loganin did not affect cellular viability in 3T3-L1 cells (Appendix A). These results indicate that loganin prevents adipocyte differentiation by reducing expressions of Pparγ, Cebpa, Plin2, Fasn, and Srebp1.

We further confirmed the anti-adipogenic effects of loganin on ADSCs isolated from mouse adipose tissues. ADSCs were induced to differentiate into adipocytes and were co-cultured with loganin (2, 5, and 10 μM) for 8 d. Consistent with the results obtained in the preadipocyte cell line, mRNA expression levels of adipogenic-related markers, including Pparγ, Cebpa, Plin2, Fasn, and Srebp1 were reduced by loganin treatment (Figure 2A), and the number of oil Red O-positive cells was also decreased (Figure 2B). These results suggest that loganin inhibits adipocyte differentiation by downregulating adipogenic-related markers (Pparγ, Cebpa, Plin2, Fasn, and Srebp1) in both 3T3-L1 mouse preadipocytes and ADSCs.

### 2.2. Loganin Prevents OVX- and HFD-Induced Weight gain in Mice

To examine the anti-adipogenic effect of loganin in vivo, we used two different animal models of obesity in mice, i.e., OVX- and HFD-induced obesity. We used the 17β-estradiol (E2; 0.03 μg/kg/d) administration as a positive control for anti-obesity, and strontium chloride (SrCl_2_; 10 mg/kg/d) administration as a negative control. E2 is a well-known reagent for treating menopausal obesity and SrCl_2_ is an anti-osteoporotic compound used for treating menopause. As expected, OVX-induced obese mice showed weight gain compared to sham-operated mice because of estrogen deficiency, further hepatic steatosis, and adipose tissue enlargement were observed. Administration of 17β-estradiol (E2), the active form of estrogen, restored OVX-induced estrogen deficiency, resulting in the prevention of weight gain, whereas the negative control group administered with the anti-osteoporotic reagent, strontium chloride (SrCl_2_), did not show any change in body weight compared to that of OVX-induced obese mice (Figure 3A). However, loganin administration prevented OVX-induced weight gain and reduced hepatic steatosis and adipose tissue enlargement (Figure 3A,B).

We further investigated the anti-adipogenic effects of loganin in a mouse model of HFD-induced obesity. Six-week-old mice were fed an HFD, and loganin treatment (2 and 10 mg/kg/d) was orally administered for 12 wk. As expected, HFD increased mouse body weight compared to the normal diet (ND) (Figure 4A), and histological analysis of the HFD-induced animals showed hepatic steatosis and adipocyte enlargement (Figure 4B). However, loganin treatment prevented HFD-induced weight gain and reduced hepatic steatosis and adipocyte expansion (Figure 4A,B). Collectively, these results suggest that loganin administration inhibits OVX- and HFD-induced weight gain, hepatic steatosis, and adipocyte enlargement.

### 2.3. Loganin Reduced Plasma Leptin and Insulin Levels in OVX- and HFD-Induced Obese Mice

Finally, we evaluated the effects of loganin on the plasma levels of leptin and insulin in OVX- and HFD-induced obese mice. OVX- and HFD-induced obese mice showed a significant increase in plasma leptin and insulin levels compared to those in the sham-operated and ND groups. However, loganin administration resulted in decreased plasma leptin and insulin levels in both OVX- and HFD-induced obese mice (Figure 5). These results indicate that loganin ameliorated the OVX- and HFD-induced increase in plasma leptin and insulin levels in mice, resulting in anti-adipogenic effects in mouse models of obesity in vivo.

## 3. Discussion

Adipogenesis promotes fat accumulation in mature adipocytes during preadipocyte differentiation, and excessive fat accumulation leads to overweightness and obesity. Regarding excessive adipogenesis initiating obesity, understanding adipocyte differentiation is important to prevent obesity-related diseases [27]. This study examined the inhibitory effects of loganin in a preadipocyte 3T3-L1 mouse cell line and in primary cultured ADSCs in vitro as well as in OVX- and HFD-induced mice in vivo.

Preadipocyte 3T3-L1 cells derived from a mouse embryonic fibroblast cell line have been widely used in biological research on adipogenesis [28]. Further, ADSCs are MSCs isolated from white adipose tissue that are most likely to recapitulate adipogenesis during adipose tissue development [29]. Complete differentiation of adipocytes is represented by the formation of lipid droplets, which are visualized using oil Red O staining [30]. In this study, 3T3-L1 preadipocytes and ADSCs induced for adipocyte differentiation and evaluated using oil Red O staining showed that loganin treatment inhibited the accumulation of lipid droplets and decreased the number of oil Red O-positive cells, indicating reduced adipocyte differentiation.

Adipocyte differentiation is regulated by various transcription factors, including *Pparγ*, *Cebpa*, *Plin2*, *Fasn*, and *Srebp1* [14,15,16]. *Pparγ* is considered to be a master regulator of adipogenesis and plays a central role in maintaining insulin sensitivity [31]. *Cebpa* binds to the *Pparγ* promoter and induces the expression of *Pparγ* isoform 2, thus enhancing adipogenesis [32]. *Plin2*, also known as an adipose differentiation-related protein, is a cytoplasmic lipid droplet-binding protein required for storing neutral lipids within lipid droplets in mature adipocytes [33,34]. Further, *Fasn* stimulates the formation of long-chain fatty acids [35,36], and *Srebp1* regulates lipogenesis and fatty acid metabolism in adipocytes [37]. In this study, we examined the mRNA expression of adipogenesis-related genes using qRT-PCR. After the induction of adipocyte differentiation, increased expression of *Pparγ, Cebpa, Plin2, Fasn,* and *Srebp1* was observed. However, loganin treatment inhibited the mRNA expression of adipogenic inducible genes in 3T3-L1 stable cells and primary ADSCs. Collectively, the in vitro results suggest that loganin treatment prevents adipocyte differentiation through the decreased accumulation of lipid droplets and downregulation of adipogenesis-related factors.

Mouse models of obesity are widely used to investigate fat development induced by HFD and OVX in mice [38,39]. The HFD contains high amounts of calories from fat and is an appropriate method to trigger excessive fat development in an in vivo obesity model [40,41]. OVX-induced obese mice lack estradiol owing to ovary removal and mimic human menopause with increased susceptibility to gain weight [39]. Based on the in vitro results, we confirmed the attenuating effects of loganin on adipogenesis in HFD- and OVX-induced obese mice. Persistent inappropriate weight gain is strongly associated with metabolic abnormalities, such as hepatic steatosis, adipocyte hypertrophy, and hyperlipidemia [42,43,44]. Liver steatosis and adipocyte enlargement are commonly reported symptoms following excessive fat deposition [45]. A recent study suggested that loganin prevented inflammatory-associated diseases by inhibiting hepatic steatosis [46]. Furthermore, excessively elevated insulin levels inhibit hormone-sensitive lipase, an essential enzyme for lipid digestion [47]. Leptin plays a major role in regulating lipid metabolism through changes in food consumption [48]. In this study, loganin treatment inhibited HFD- and OVX-induced weight gain and fat deposition reduced metabolic abnormalities, such as hepatic steatosis and adipocyte expansion, and increased the plasma levels of insulin and leptin. The results indicated that the protective effects of loganin on metabolic abnormalities induced by HFD and OVX are probably due to anti-obesity effects rather than phytoestrogen activity. Our results thus showed that loganin reduced the total body weight along with adipogenic-associated abnormalities in two mouse models of obesity.

Collectively, loganin promoted the reduction of adipocyte differentiation and accumulation of lipid droplets in 3T3-L1 preadipocytes and ADSCs and alleviated obesity-related phenotypes induced by OVX and HFD in vivo.

## 4. Materials and Methods

### 4.1. Reagents, Cell Culture and Induction of Mature Adipocytes

Loganin was purchased from Chengdu Biopurify Phytochemicals Ltd., (Sichuan, China) and was completely dissolved in deionized water. The mouse fibroblast cell line, 3T3-L1, was obtained from the Korean Cell Line Bank (KCLB No. 10092.1). 3T3-L1 cells were maintained in high-glucose Dulbecco’s modified Eagle’s medium (DMEM; Invitrogen, Carlsbad, CA, USA) containing 10% bovine calf serum (BCS; Invitrogen, Carlsbad, CA, USA) and 1% antibiotic-antimycotic (AA; Invitrogen, Carlsbad, CA, USA). For adipogenic induction, cells (1×10^6^ cells) were seeded in 6-well plates (SPL Life Sciences, Pocheon, Republic of Korea) and maintained until the cells reached 100% confluent. Then, the cells were replaced with DMEM containing 10% fetal bovine serum (FBS; Invitrogen, Carlsbad, CA, USA), 1% AA, 1 μM dexamethasone, 0.5 mM 3-isobutyl-1-methylxanthine, and 10 μg/mL insulin for 3 days. The medium was then incubated with DMEM containing 10% FBS, 1% AA, and 10 μg/mL insulin for 5 days. Insulin was changed every 2 days, and loganin was replaced every time the media was switched. ADSCs were isolated using the stromal vascular fraction, as previously described [49]. Briefly, 9-week-old mouse epidydimal adipose tissue was digested with collagenase type II for 1 h. The digestive solution was neutralized with low-glucose DMEM containing 10% FBS, followed by filtration using a 100 μm cell strainer (Corning, NY, USA). The cells were then centrifuged at 2500 rpm for 10 min and maintained in low-glucose DMEM containing 10% FBS and 1% AA. For the adipogenic induction of ADSCs, cells were incubated with Mesencult^™^ Adipogenic Differentiation Medium (STEMCELL Technologies, Vancouver, BC, Canada) for 8 d. The “Control” indicates non-treated cells, and the “Mock” indicates adipogenic induction medium-treated cells. To examine cellular viability tests, 3T3-L1 cells were incubated with loganin in cultured media for 8 d and cellular viability was assessed using D-Plus^™^ CCK cell viability kit (Dongin Biotech, Seoul, Republic of Korea) in absorbance at 450 nm by iMark^™^ Microplate Absorbance Reader (Bio-Rad, Hercules, CA, USA).

### 4.2. Oil Red O Staining

The cells were fixed with 4% paraformaldehyde (BIOSESANG, Seongnam, Republic of Korea) for 15 min and then with 70% isopropanol for 1 min. Thereafter, the cells were incubated with oil Red O (Sigma-Aldrich, St. Louis, MO, USA) for 1 h. Representative images were obtained using a light microscope (Leica Microsystems; Wetzlar, Germany). For quantification of oil Red O-positive cells, cells were destained with 100% isopropanol, and absorbance at 490 nm was measured using a microplate reader (Bio-Rad, Hercules, CA, USA). The values were normalized to the “Mock” sample (1.0) and expressed as relative values for the other samples.

### 4.3. Quantitative Reverse Transcription Polymerase Chain Reaction (qRT-PCR)

Total RNA was isolated using the QIAzol Lysis Reagent (QIAGEN, Hilden, Germany), according to the manufacturer’s instructions. RNA was reverse-transcribed using the RevertAid^™^ H Minus First Strand cDNA synthesis kit (Fermentas, Hanover, NH, USA) under the following conditions: 2 U of Dnase Ⅰ for 30 min at 37 °C, 50 mM EDTA for 10 min at 65 °C, 1:1 ratio of Random Hexamer and Oilgo (dT) 18 primers for 5 min at 65 °C and 10 mM of dNTP mix, 20 U of RNase Inhibitor, and 200 U of RevertAid H Minus Reverse Transcriptase for 5 min at 25 °C, 1 h at 42 °C and 5 min at 70 °C. qRT-PCR was performed using the SYBR Green I qPCR kit (Takara, Shiga, Japan). The gene-specific primers used in this study were as follows: forward 5′-GCG GGA ACG CAA CAA CAT C-3′ and reverse 5′-GTC ACT GGT CAA CTC CAG CAC-3′ for mouse *Cebpa*, forward 5′-AAG ATG TAC CCG TCC GTG TC-3′ and reverse 5′-TGA AGG CAG GCT CGA GTA AC-3′ for mouse *Srebp1*, forward 5′-GGA AGA CCA CTC GCA TTC CTT-3′ and reverse 5′-GTA ATC AGC AAC CAT TGG GTC-3′ for mouse *Pparg*, forward 5′-GAC CTT GTG TCC TCC GCT TAT-3′ and reverse 5′-CAA CCG CAA TTT GTG GCT C-3′ for mouse *Plin2*, forward 5′-GGA GGT GGT GAT AGC CGG TAT-3′ and reverse 5′-TGG GTA ATC CAT AGA GCC CAG-3′ for mouse *Fasn*, and forward 5′-AGC TGA AGC AAA GGA AGA GTC GGA-3′ and reverse 5′-ACT TGG TTG CTT TGG CGG GAT TAG-3′ for mouse *Arbp*. Relative mRNA expression levels were normalized to those of mouse *Arbp* (ribosomal protein large P0, also known as *Rplp0*) expression, and fold change was determined using the 2^−ΔΔCt^ method. The values presented in this study were expressed using “Mock” as a standard (1.0), while other values were expressed as relative values.

### 4.4. Animal Study

All animal experiments performed in this study were approved by the Institutional Animal Care and Use Committee (IACUC) of Ajou University School of Medicine (2022-0064). Mice were maintained under specific-pathogen-free conditions at the Animal Care Center at Ajou University School of Medicine and provided with standard food pellets (Feedlab Co., Ltd., Hanam, Republic of Korea) and distilled water *ad libitum*. The OVX- or HFD-induced obese mice were used as previously described [50,51]. For OVX-induced obese mice, sham-operated (n = 5) and OVX-induced ddY mice (n = 25) were purchased from Shizuoka Laboratory Center Inc. (Hamamatsu, Japan). OVX-induced obese mice were divided into five groups: OVX only, OVX plus β-estradiol (E2; 0.03 μg/kg/day, Sigma-Aldrich), OVX plus strontium chloride (SrCl_2_; 10 mg/kg/day, Sigma-Aldrich), OVX plus loganin (2 mg/kg/day), and OVX plus loganin (10 mg/kg/day). For HFD-induced obese mice, 4-week-old mice were divided into four groups (n = 5 per group): ND, HFD, HFD plus loganin (2 mg/kg/day), and HFD plus loganin (10 mg/kg/day). The total body weights of the mice were measured using a Micro Weighing Scale (CAS Corporation, Yangju, Republic of Korea) after 4, 8, and 12 weeks of the experiment. E2, SrCl_2_, and loganin were administered through oral gavage. At the end of the experiment, mice were euthanized using CO_2,_ and tissue samples, including liver and fat, were fixed in 4% paraformaldehyde (BIOSESANG, Seongnam, Republic of Korea).

### 4.5. Histological Analysis

Formalin-fixed tissue samples were dehydrated and embedded in paraffin. The paraffin blocks were sectioned using a rotary microtome (3 μm; Leica Microsystems, Wetzler, Germany). The tissue slides were stained with hematoxylin and eosin (H&E; SSN Solutions, London, UK). Briefly, the sectioned slides were deparaffinized using xylene and rehydrated using sequentially treated ethanol (100%, 95%, and 70%). Slides were stained with Harris hematoxylin solution and differentiated using 1% acid alcohol. Bluing was performed using 0.2% ammonia water and counterstained with eosin Y solution. The slides were then dehydrated using sequentially treated ethanol (70%, 95%, and 100%), cleared with xylene, and mounted using mounting medium (Leica Microsystems, Wetzler, Germany). Slide scanning was performed using an Axioscan Z1 slide scanner (Carl Zeiss).

### 4.6. Plasma Analysis

At the end of the experiment, blood samples were obtained from mice using cardiac puncture, collected in EDTA tubes, and stored at −80 °C until use. Plasma leptin and insulin levels were determined using a customized MILLIPLEX^®^ Mouse Adipokine Magnetic Bead Panel (MADKMAG-71K; Millipore, Billerica, MA, USA) and a MAGPIX^®^ multiplex analyzer (Luminex, Austin, TX, USA).

### 4.7. Statistical Analysis

Data in bar graphs are expressed as mean ± standard error of the mean (SEM) using GraphPad Prism 9.2.0 software (GraphPad Software, San Diego, CA, USA). Statistical significance was determined using one-way analysis of variance (ANOVA), followed by Tukey’s honest post hoc test using the professional Statistical Package software (SPSS 25.0 for Windows, SPSS Inc., Chicago, IL, USA).

## 5. Conclusions

This study revealed the inhibitory effects of loganin on adipogenesis in 3T3-L1 preadipocytes, ADSCs, and on OVX- and HFD-induced obesity models in mice. Loganin treatment decreased adipocyte differentiation and lipid droplet accumulation by reducing the mRNA expression of adipogenesis-related factors. In OVX- and HFD-induced obese mice, loganin attenuated the representative obesity phenotypes, including hepatic steatosis, adipocyte hypertrophy, and increased plasma levels of leptin and insulin. These findings indicate the strong potential of loganin as a therapeutic agent for treating and preventing obesity.

## Figures and Tables

**Figure 1 ijms-24-04752-f001:**
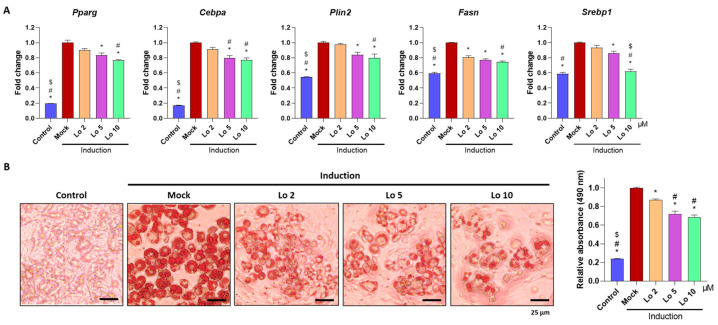
Inhibitory effects of loganin on mouse preadipocyte differentiation. Notably, 3T3-L1 cells underwent adipogenesis after being treated with MDI (3-isobutyl-1-methylxanthine, dexamethasone, and insulin) for 3 days, followed by insulin treatment for 5 days. During the 8 days of the adipogenesis period, loganin was administered at various concentrations (2, 5, and 10 μM). (**A**) mRNA expression levels of Pparg, Cebpa, Plin2, Fasn, and Srebp1 were examined using qRT-PCR. (**B**) Oil Red O-positive cells were visualized using a light microscope (**left**), and stained cells were quantified by relative absorbance at 490 nm (**right**) using microplate reader. * *p* < 0.05 vs. Induction, ^#^
*p* < 0.05 vs. Lo 2. ^$^
*p* < 0.05 vs. Lo 5. Lo; Loganin.

**Figure 2 ijms-24-04752-f002:**
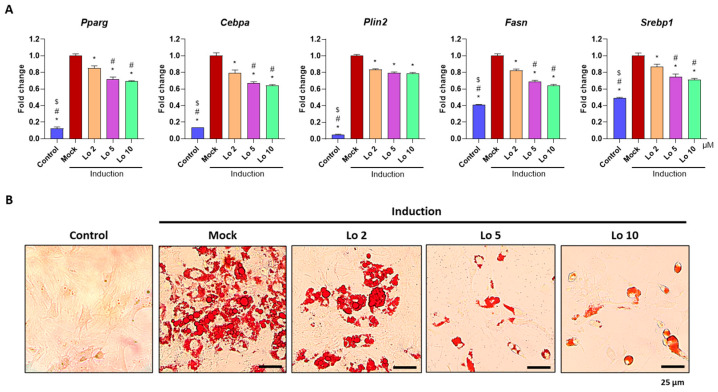
Inhibitory effects of loganin on mouse primary adipocyte differentiation. ADSCs were induced for adipocyte differentiation and co-treated with loganin (2, 5, and 10 μM) for 8 d. (**A**) mRNA expression levels of Pparg, Cebpa, Plin2, Fasn and Srebp1 were examined using qRT-PCR. (**B**) Oil Red O-positive cells were visualized using a light microscope. * *p* < 0.05 vs. Induction, ^#^
*p* < 0.05 vs. Lo 2, ^$^
*p* < 0.05 vs. Lo 5. Lo, Loganin.

**Figure 3 ijms-24-04752-f003:**
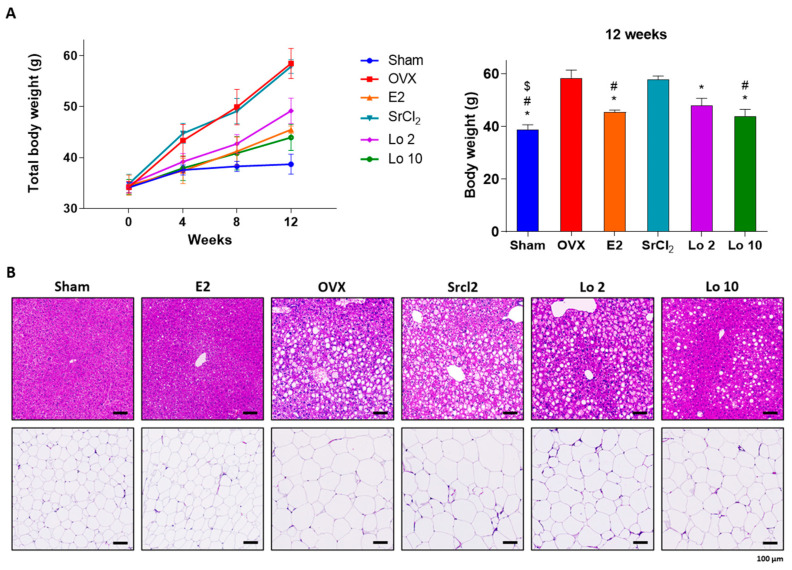
Ameliorative effects of loganin on OVX-induced mouse gain weight. OVX-induced obese mice were administered E2, SrCl_2_, or loganin (2 and 10 mg/kg/day) for 12 weeks. (**A**) Body weight changes during the 12 weeks of the experiment. n = 5 per group. (**B**) Representative images of H&E-stained mouse liver and fat tissue sections. * *p* < 0.05 vs. OVX, ^#^
*p* < 0.05 vs. SrCl_2_, ^$^
*p* < 0.05 vs. Lo 2. Sham, sham-operated mice; OVX, ovariectomized mice; Lo, loganin.

**Figure 4 ijms-24-04752-f004:**
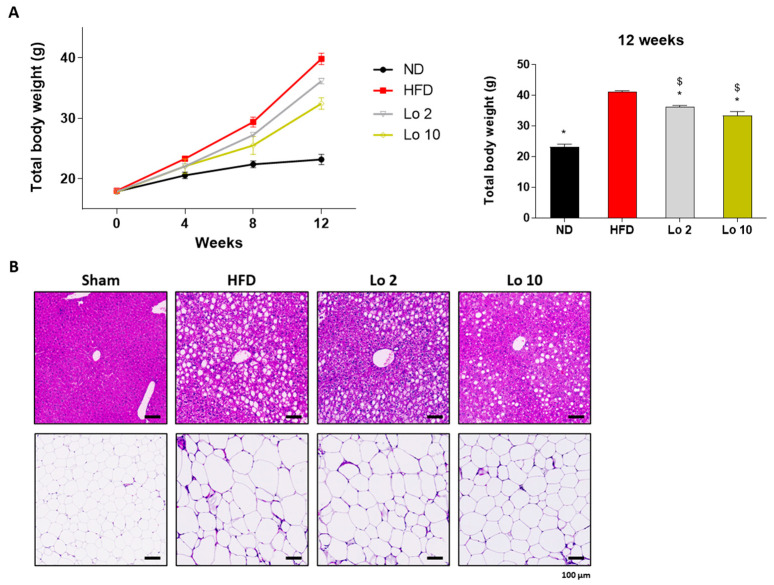
Anti-obesity effects of loganin on HFD-induced obese mice. Mice were administered with ND, HFD, or HFD with different concentrations of loganin (2 and 10 mg/kg) for 12 weeks. (**A**) Body weight changes throughout 12 weeks of the experiment. n = 5 per group. (**B**) Representative images of H&E-stained mouse liver and fat tissue sections. * *p* < 0.05 vs. HFD, ^$^
*p* < 0.05 vs. Lo 2. ND, normal diet; HFD, high-fat diet; Lo, loganin.

**Figure 5 ijms-24-04752-f005:**
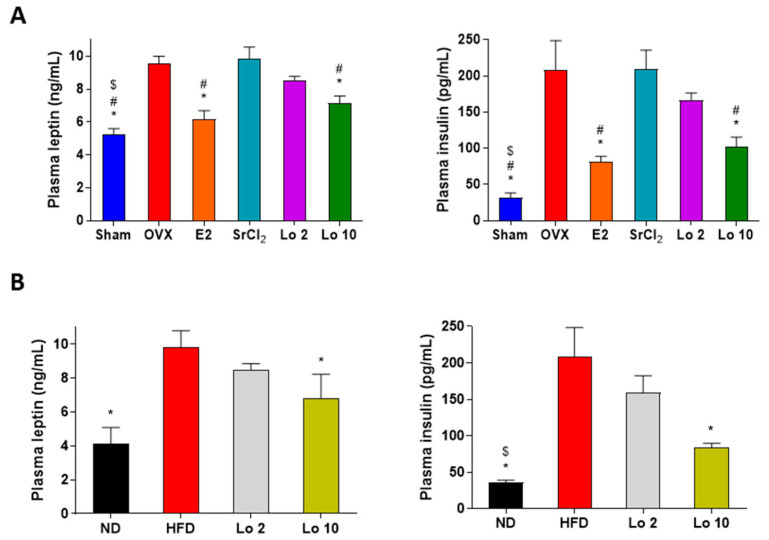
Effects of loganin on plasma levels of leptin and insulin in obese mice. (**A**) OVX-induced obese mice were administered E2, SrCl_2_, or loganin (2 and 10 mg/kg/day) for 12 weeks. (**B**) Mice were administered with ND, HFD, or HFD with different concentrations of loganin (2 and 10 mg/kg) for 12 weeks. Plasma leptin (**left**) and insulin (**right**) levels were examined using ELISA. * *p* < 0.05 vs. OVX or HFD, ^#^
*p* < 0.05 vs. SrCl_2_, ^$^
*p* < 0.05 vs. Lo 2. Sham, sham-operated mice; OVX, ovariectomized mice; ND, normal diet; HFD, high-fat diet; Lo, loganin.

## Data Availability

The data presented in this research is available on request from the corresponding author.

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
