# Peer review of "Inhibitory Effects of Loganin on Adipogenesis In Vitro and In Vivo"

_ijms, 2023, doi:10.3390/ijms24054752_

Round 1

Reviewer 1 Report

This study is very interesting. Some of the explanation of results seems slightly rough. More detailed explanation is preferable.

Author Response

Best regards,

Reviewer 2 Report

Manuscript review response: “Inhibitory effects of loganin on adipogenesis in vitro and in vivo”.

Manuscript ID: ijms-2159579

Comments

It is an article that could provide relevant information about the anti-adipogenic effect in vitro and in vivo of this interesting compound, but is necessary there are some changes, add some inscriptions and put some bibliographic citations.

Line 21-22 could you add the techniques and methods used in this research?

Line 43-44 could you mentioned what kind of inflammation is promoted by the obesity and what kind of liver damage?

Line 45 It is necessary to add the mechanism by which cells cause insulin resistance and what type of cells carry out these mechanisms.

Line 47 you could describe the mechanism by which leptin resistance is generated.

Line 55-56 It is important to mention and describe the hypertrophic or hyperplastic processes that are observed in obesity.

Line 57 Could you mention the half-life of leptin, insulin and adiponectin in plasma? and add it to the writing.

Line 58 you need mentioned what kind of eating behaviour is caused by obesity.

Line 59-60 What are the pharmacological therapies to reduce obesity and what are their serious adverse effects and long-term safety limitations?

Line 70 It is important to mention the characteristics of loganin and what type of compound it is, considering the antecedents that have been reported.

Line 110 Was a single dose of loganin administered in 8 days or were they daily doses during the 8 days?

Line 150 In Figure 4a, the OVX model that shows the decrease in weight is not present.

Line 214 in this paragraph you mentioned that “continuous high glucose levels caused by the HFD and OVX promote 214 abnormal insulin and leptin secretion with alteration of hepatic fat regulation” did your animal model measure blood glucose levels? What is the relevance of this evidence?

Line 216-223 these paragraphs are from the material and methods section please change “To evaluate the effects of loganin on the OVX-induced obesity mouse model, 8-week-old female mice were either sham-operated or OVX and divided into six groups: (1) sham, (2) OVX, (3) OVX administrated with 17β-estradiol (E2; 0.03 μg/kg/day) as a positive control for anti-obesity, (4) OVX administrated with strontium chloride (SrCl2; 10 mg/kg/day) as a negative control, (5) OVX administrated with 2 mg/kg/day of loganin, and (6) OVX administrated with 10 mg/kg/day of loganin for 12 weeks. E2 is an active form of estrogen for the treatment of menopausal obese mouse model and SrCl2 is used as an anti-osteoporotic compound for 123 the treatment of menopause. E2, SrCl2, and loganin were administered by oral gavage.

Line 230 you mentioned that 3T3-L1 cell lines were pre-adipocytes, but this line are fibroblasts, the affirmation is incorrect, you need change this paragraph.

Line 239 What confluence did the cell culture reach?

Line 246 could you add experimental design of cultured in well plates (6 well plates, 24 well plates) what kind of plates did you used?

247 how many cells did you use per well, you need add this information in methods.

Line 249 the final evaluation of oily red was 1 hour? did you get 24-hour results?

Line 254 could you add synthesis conditions for cDNA obtention?

Line 268 It is important to cite the ovariectomized animal model or describe the technique for developing the model in mice and HFD-induced mice in vivo.

Line 284 Does the use of CO2 for euthanasia in animals affect the results in your analysis?

Line 289 the haematoxylin and eosin technique need to be cited or described in this section.

Line 156 How could you show that there is resistance to insulin and leptin? In the results obtained, did you observe resistance to any of these proteins?

Author Response

Best regards,

Reviewer 3 Report

The authors of this paper have evaluated the potential regulatory role of Loganin on adipogenesis through both in vitro and in vivo assays. The results suggest that the treatment of Loganin may inhibit adipogenesis by decreasing the expression of adipogenic transcription factors, including PPARγ and C/EBPa. This study could be a valuable resource for the field. However, there are some issues that need to be addressed before the paper can be considered for publication.

Main concerns:

The authors performed qPCR analysis on multiple genes involved in adipogenic regulation in their in vitro studies. The rationale for gene selection is unclear. PPARγ and C/EBPα are two key transcription factors that regulate adipogenesis, while Fasn and SREBP1 are upstream activators for either PPARγ or C/EBPα. Plin1 or Plin2 encode proteins expressed on lipid droplets and are used as markers for mature adipocytes. These genes cannot be grouped together and referred to as "adipogenic markers" as the authors did in the manuscript. The decreased transcription of these genes has different implications. For example, it's incorrect to say "Loganin inhibits adipocyte differentiation by downregulating Plin2," as Plin2 is a marker for mature adipocytes but does not regulate adipocyte differentiation. The authors need to provide a clearer explanation and interpretation of their results.

In their in vivo adipogenesis model, the authors need to specify whether the morphological analysis was performed on subcutaneous or visceral WAT. This is because OVX may induce adipogenesis differently in these two types of WAT. It is also recommended to provide H&E staining from the interscapular BAT in both models.

Minor issue:

The logic of the paper is somewhat difficult to follow due to poor writing quality. Grammatical errors are present throughout the manuscript. For example, the sentence "Loganin treatment reduced adipocyte differentiation by accumulating lipid droplets through..." (line 22) does not make sense. The authors should perform a thorough proofread before submitting the paper again.

Author Response

Best regards,

Round 2

Reviewer 2 Report

I don't have any questions

Reviewer 3 Report

None